# Cervicovaginal Microbiota Profiles in Precancerous Lesions and Cervical Cancer among Ethiopian Women

**DOI:** 10.3390/microorganisms11040833

**Published:** 2023-03-24

**Authors:** Brhanu Teka, Kyoko Yoshida-Court, Ededia Firdawoke, Zewditu Chanyalew, Muluken Gizaw, Adamu Addissie, Adane Mihret, Lauren E. Colbert, Tatiana Cisneros Napravnik, Molly B. El Alam, Erica J. Lynn, Melissa Mezzari, Jhingran Anuja, Eva Johanna Kantelhardt, Andreas M. Kaufmann, Ann H. Klopp, Tamrat Abebe

**Affiliations:** 1Department of Microbiology, Immunology and Parasitology School of Medicine, College of Health Sciences, Addis Ababa University, Addis Ababa P.O. Box 9086, Ethiopia; 2Global Health Working Group, Martin-Luther-University, Halle-Wittenberg, 06097 Halle, Germany; 3Department of Radiation Oncology, The University of Texas MD Anderson Cancer Center, Houston, TX 77030, USA; 4Department of Pathology, St. Paul Hospital Millennium Medical College, Addis Ababa P.O. Box 1271, Ethiopia; 5School of Public Health, College of Health Sciences, Addis Ababa University, Addis Ababa P.O. Box 34, Ethiopia; 6Institute for Medical Epidemiology, Biometrics and Informatics, Martin-Luther-University, Halle-Wittenberg, 06120 Halle, Germany; 7NCD Working Group, Addis Ababa University, Addis Ababa P.O. Box 34, Ethiopia; 8Armauer Hansen Research Institute, Addis Ababa P.O. Box 1005, Ethiopia; 9Molecular Virology and Microbiology, Baylor College of Medicine Alkek, Center for Molecular Discovery, Houston, TX 77030, USA; 10Department of Gynecology, Charité–Universitätsmedizin Berlin, Corporate Member of Freie Universität Berlin and Humboldt-Universität zu Berlin, 10117 Berlin, Germany

**Keywords:** Ethiopia, Tikur Anbessa Specialized Hospital, high-risk HPV, cervical intraepithelial neoplasia, cervical microbiota

## Abstract

Although high-risk human papillomavirus infection is a well-established risk factor for cervical cancer, other co-factors within the local microenvironment may play an important role in the development of cervical cancer. The current study aimed to characterize the cervicovaginal microbiota in women with premalignant dysplasia or invasive cervical cancer compared with that of healthy women. The study comprised 120 Ethiopian women (60 cervical cancer patients who had not received any treatment, 25 patients with premalignant dysplasia, and 35 healthy women). Cervicovaginal specimens were collected using either an Isohelix DNA buccal swab or an Evalyn brush, and ribosomal RNA sequencing was used to characterize the cervicovaginal microbiota. Shannon and Simpson diversity indices were used to evaluate alpha diversity. Beta diversity was examined using principal coordinate analysis of weighted UniFrac distances. Alpha diversity was significantly higher in patients with cervical cancer than in patients with dysplasia and in healthy women (*p* < 0.01). Beta diversity was also significantly different in cervical cancer patients compared with the other groups (weighted UniFrac Bray-Curtis, *p* < 0.01). Microbiota composition differed between the dysplasia and cervical cancer groups. *Lactobacillus iners* was particularly enriched in patients with cancer, and a high relative abundance of *Lactobacillus* species was identified in the dysplasia and healthy groups, whereas *Porphyromonas*, *Prevotella*, *Bacteroides*, and *Anaerococcus* species predominated in the cervical cancer group. In summary, we identified differences in cervicovaginal microbiota diversity, composition, and relative abundance between women with cervical cancer, women with dysplasia, and healthy women. Additional studies need to be carried out in Ethiopia and other regions to control for variation in sample collection.

## 1. Introduction

Cervical cancer is a major public health problem and is the fourth most common cancer among women globally [1]. In 2020, an estimated 604,237 women globally had cervical cancer, representing 6.5% of all female cancers. About 90% of the new cases and deaths worldwide in 2020 occurred in low- and middle-income countries [2]. In Ethiopia, cervical cancer is the second most common cancer with 6300 annual new cases and about 4884 deaths each year [3].

It is well established that persistent infection with high-risk human papilloma virus (HPV) genotypes causes high-grade cervical intraepithelial neoplasia (CIN2+) and invasive cervical cancer [4]. However, HPV infection alone is insufficient to cause cancer; most genital HPV infections are transient and asymptomatic. Only a small number of women—those chronically carrying oncogenic or high-risk HPV genotypes—eventually develop severe dysplasia (CIN2/3) and, over several decades, cancer [5,6]. Therefore, the most important aspect in cervical cancer prevention is to focus on precancerous lesions and categorize those women according to their risk of persistence or recurrence so as to identify the women at higher risk [7,8].

Mechanisms associated with clearance or persistence of HPV infection are not well understood. Currently, different factors have been found to be associated with persistent HPV infection and its development into cervical dysplasia and/or invasive cancer [8]. These include, architecture of the epithelial surface, mucosal immunity, other sexually transmitted infections, and the cervicovaginal microbiota and microenvironment [9,10]. The prevalence of any HPV or multiple HPV infections is higher among human immunodeficiency virus (HIV) positive women than HIV negative women in sub-Saharan Africa [11]. Lactic acid–producing bacteria and mucosal secretions have been shown to play an important role in trapping pathogens, including HPV, HIV, and others that cause sexually transmitted infections [12,13,14]. The protective role of these lactic acid–producing bacteria, i.e., *Lactobacillus* species, is disrupted following development of highly diverse vaginal microorganisms associated with proinflammatory responses. The inflammatory response may in turn cause direct damage to the epithelium and facilitate HPV entry, as well as persistent infection [13,15,16].

Although more than 50 microbial species have been identified in the vaginal tract, *Lactobacillus* species, mainly *L. crispatus*, *L. gasseri*, *L. iners*, and *L. jensenii*, make up about 70% of the vaginal microbiota [17]. The structure and function of the vaginal microbiota can vary with genetic disposition, ethnicity, diet, hygiene, infections, antibiotic use, sexual activity, physiologic status of the vagina, and especially estrogen levels [17,18]. Studies have indicated that increased CIN grade is associated with increased vaginal microbiota diversity, which may be involved in regulating viral persistence and disease progression [19]. However, specific microbes or microbial communities that can be mechanistically linked to cervical carcinogenesis remain largely unidentified [20]. Recent studies revealed that decreased levels of probiotics, such as *Shuttleworthia*, *Prevotella*, *Lactobacillus*, and *Sneathia*, along with high levels of pathogenic bacteria, including *Dispar*, *Streptococcus*, and *Faecalibacterium prausnitzii*, could be the direct result of early HPV infection. In addition, some pathogenic bacteria, such as *Bifidobacteriaceae*, could be involved in cancer progression [21].

With the current rapidly progressing sequencing technologies that characterize microbial communities beyond culture-based or biochemical techniques, cervicovaginal microbiota sequencing and analysis, most specifically 16S ribosomal RNA (rRNA) amplicon sequencing, is becoming the best tool to advance knowledge of the association between microbiota and cervical cancer progression [22]. 16S rRNA gene sequencing produces reliable taxonomic classifications and relative abundances and is a cost-effective method to quantify diversity metrics, as well as provide genus-level identification [23]. However, 16S rRNA sequencing has some disadvantages, including the inability to provide species-level identification and metagenomic functionality. Furthermore, this method includes only bacteria; the virome or other non-bacterial members of the microbiota cannot be identified without targeted screening [24]. These deficiencies of 16S rRNA sequencing can be overcome with whole-genome shotgun sequencing [25].

Recently we followed a community of women who attended HPV-based cervical cancer screening for 2 years and learned that some cleared the virus within 6 months whereas others had persistent infections. In an effort to understand the possible underlying factors, we characterized the cervicovaginal microbiota structure and diversity in women with cervical cancer, women with dysplasia (various CIN grades), and healthy women in central and south-central Ethiopia. 

## 2. Materials and Methods

### 2.1. Study Area

The study was carried out at Tikur Anbessa Specialized Hospital, Addis Ababa, Ethiopia, and in the rural community of Butajira in south central Ethiopia (Gurage Zone of the Southern Nations, Nationalities, and Peoples’ Region of Ethiopia). The gynecology department at the hospital provided evaluation, examination, surgical treatment, and screening of new and referred cases of cervical cancer. 

### 2.2. Inclusion and Exclusion Criteria

At Tikur Anbessa Specialized Hospital, all eligible women aged 18 years or older who were attending the cervical cancer screening program from October 2019 to February 2020 were screened, and those with histologically confirmed cervical cancer or precancerous lesions were included in the current study. In Butajira, women who underwent cytologic examination for cervical cancer screening and whose results were either normal or cytologically positive for precancerous lesions were included in the current study. Women who were pregnant; had undergone hysterectomy, chemotherapy, or radiotherapy; or had a cognitive or physical impairment that prevented them from giving informed consent or participating were excluded from the study.

This comparative study comprised 120 women: (60 who had cervical cancer and had not received any treatment, 25 who had cytologically or histologically confirmed cervical dysplasia, and 35 who were cytologically or histologically negative for cervical dysplasia (Table 1).

### 2.3. Sample Collection and DNA Extraction

Specimens for the current study were collected using one of two devices: 67 specimens were collected using an Isohelix DNA buccal swab, and 53 specimens were collected using an Evalyn brush (Rovers Medical Devices, Oss, The Netherlands). Among the samples collected using Isohelix swabs, 60 were cancer, 6 were dysplasia, and 1 was negative for intraepithelial lesions or malignancy; among those collected using the Evalyn brush, 0 were cancer, 19 were dysplasia, and 34 were negative (Appendix A).

The Isohelix cervical swab specimens were obtained by first swabbing the surface of the cervix and then placing the swab into a stabilization buffer, followed by freezing within 30 min at −20 °C. The Evalyn brush is a self-collection device, and women were assisted by trained health workers for sample collection. The brush was placed in a plastic bag and stored in a dry place at room temperature until it could be transported to the HPV laboratory of the Department of Microbiology, Immunology and Parasitology, School of Medicine, Addis Ababa University, for processing.

The Evalyn brushes were processed as previously described [26]. Briefly, the brush was removed from the plastic bag and the tip was pulled off and placed into a 2-mL Eppendorf tube and soaked in 1 mL phosphate-buffered saline overnight to remove the cells from the dry brush. The tube was then centrifuged for 5 min at 2500 rpm and vortexed vigorously for 1 min, and an aliquot of 200 μL of the fluid was used for DNA extraction. DNA extraction was carried out using the Qiagen DNA Extraction Kit (QIAGEN, Hilden, Germany). Both the Isohelix and Evalyn brush samples were shipped to The University of Texas MD Anderson Cancer Center, USA, on dry ice for DNA extraction and 16S rRNA sequencing.

### 2.4. HPV Genotyping

HPV presence and genotype were determined using the Anyplex II HPV 28 detection kit (Seegene, Seoul, Korea), which uses real-time multiplex PCR with tagging oligonucleotide cleavage and extension technology for simultaneous detection and genotyping of high-risk and low-risk HPV genotypes. Anyplex can detect 28 HPV genotypes including 19 high-risk types (16, 18, 26, 31, 33, 35, 39, 45, 51, 52, 53, 56, 58, 59, 66, 68, 69, 73, 82) and 9 low-risk types (6, 11, 40, 42, 43, 44, 54, 61, 70).

### 2.5. 16S rRNA Gene Sequencing and Sequence Data Processing

16S rRNA gene sequencing was performed by the Alkek Center for Metagenomics and Microbiome Research at Baylor College of Medicine (Houston, TX, USA). Sequencing was performed based on methods adapted from the Human Microbiome [27]. DNA was extracted using the MO BIO PowerSoil DNA Isolation Kit (MO BIO Laboratories, Carlsbad, CA USA). The 16S V4 region is the most conserved and variable segment of the genome, which makes it a good target for phylogenetic analyses. This region was amplified by PCR using a 515F-806R primer pair. Sequencing was performed on the Illumina MiSeq platform using the 2 × 250 base pair paired-end protocol, yielding pair-end reads.

16S rRNA sequence reads were processed using the QIIME2 microbiome bioinformatics platform (v2020.11) [28]. FASTQ sequences were imported and demultiplexed as QIIME2 artifacts. Denoising was performed using DADA2 with trim parameters set at 20–245 for forward strands and 8–230 for reverse strands. Trim parameters were set based on quality score plots generated on QIIME2 [29]. Representative sequences were generated using DADA2 for phylogenetic tree construction and taxonomic classification. A pre-trained naïve Bayes classifier was used for phylogenetic reference construction via the q2-feature-classifier plugin. We used a taxonomic classifier trained on the SILVA 138 database 515F/806R region of sequences trimmed to include 250 bases from the 16S V4 region [30].

### 2.6. Statistical Analyses

The Shannon diversity index and Simpson index were used to assess a within-sample (alpha) diversity-based OTU table in QIIME, and results were compared among multiple groups (Kruskal test) or between two groups (pairwise Wilcoxon test). In addition, beta diversity analysis was performed using UniFrac distance metrics and visualized via principal coordinate analysis.

### 2.7. Ethical Considerations

The study was approved by the Institutional Review Board of the College of Health Sciences at Addis Ababa University and by the National Research Ethics Review Committee of the Ministry of Science and Higher Education-Ethiopia. A material transfer agreement was signed by both institutions to transfer samples to MD Anderson for processing.

## 3. Results

### 3.1. Participant Characteristics

The general characteristics of the 120 participants (median age = 40 years) are provided in Table 1. The study included 35 women with normal cytologic or histologic characteristics, 25 with low-grade or high-grade dysplasia, and 60 with cervical cancer (either squamous cell carcinoma or adenocarcinoma). Most participants (n = 84, 70%) were younger than 50 years; 35 (29%) were aged 50 years or older. Twenty-seven women (23%) were HPV-negative and 93 (78%) were HPV-positive (any high-risk HPV genotype). Among those younger than 50 years (n = 84), 28 had cervical cancer, 22 had dysplasia, and 34 had normal cytologic or histologic characteristics. Among those aged 50 years or older, 32 had cervical cancer, 2 had dysplasia, and 1 had normal cytologic or histologic characteristics (Appendix A). No participants in our cohort had a low-risk HPV genotype. Cervical dysplasia was classified according to histologic grade of cervical intraepithelial neoplasia (CIN1–3) or according to cytologic grading of squamous intraepithelial lesions (high-grade or low-grade).

### 3.2. Age-Related Composition and Diversity Changes in Cervicovaginal Microbiota

We observed age-associated alterations in cervical microbiota composition and diversity. Alpha diversity significantly increased with age (Shannon *p* = 0.00071, Simpson *p* = 0.0041) (Figure 1A). Those younger than 50 years (n = 84) had significantly increased levels of *Lactobacillus* (*p* = 0.00003) and *Gardenella* (*p* = 0.01), and those aged 50 years or older (n = 35) had increased levels of *Porphyromonas* (*p* = 0.00006), *Prevotella* (*p* = 0.0041), *Bacteroids* (*p* = 0.0023), and *Anaerococcus* (*p* = 0.00097; Figure 1B). However, because the two age groups had an uneven distribution of histologic or cytologic characteristics (Appendix A), we compared only cancer patients from each age group (<50 years, n = 28; and ≥50 years, n = 32). There were no differences in alpha (Shannon *p* = 0.32, Simpson *p* = 0.39) or beta (*p* = 0.46) diversities or microbiota composition among the women with cancer (Appendix A).

### 3.3. Composition and Diversity of Cervicovaginal Microbiota among HPV-Positive and HPV-Negative Women

In our study population, we compared the alpha and beta diversity between women who were HPV-negative and those who were HPV-positive. There was no significant difference in alpha diversity between HPV- positive and HPV- negative women (Shannon *p* = 0.21, Simpson *p*= 0.18; Appendix A). Principal coordinate analysis ((weighted UniFrac, Bray-Curtis) revealed significant differences in microbial diversity between HPV- positive and HPV- negative women (*p* = 0.002; Appendix A).

Next, we compared the composition of the most abundant genera and species between HPV- positive and HPV-negative women. Among the top 10 most abundant genera, *Porphyromonas* (*p* = 0.0204) and *Peptoniphilus* (*p* = 0.0423) were significantly more abundant in the HPV- positive group (Appendix A). However, we observed no significant differences in species abundance between HPV-positive and HPV-negative women (Appendix A).

### 3.4. Cervicovaginal Microbiota Diversity and Composition among Healthy Women, Women with Cervical Dysplasia, and Women with Cervical Cancer

We compared cervicovaginal microbiota diversity among healthy women, women with dysplasia, and women with cervical cancer. We observed a significant difference in richness and evenness among the three groups (Shannon *p*= 0.00000054, Simpson *p*= 0.000005; Figure 2A). In pairwise comparisons, all indices showed that the cervical cancer group had the highest community diversity of cervicovaginal microbiota (*p* < 0.05, Kruskal-Wallis test). Principal coordinate analysis of the Bray-Curtis distances revealed differences in community structure in the cervical cancer samples compared with dysplasia and negative samples (weighted Bray-Curtis UniFrac, *p* = 0.001; Figure 2B).

Next, we characterized the microbiota taxa abundances in our cohort. A stacked bar plot (Figure 3) showed distinct patterns of taxa in our cohort. *Lactobacillus* was the dominant genus.

The relative abundance of cervicovaginal microbiota at the genus and species level was also characterized. The top 10 most abundant genera in the samples varied by group. A heat map (Figure 4) reflected the relative abundance of the most prevalent bacterial genera, showing that cervicovaginal microbiota composition changed from less diverse and more Lactobacillus-dominant in dysplasia to more diverse and less *Lactobacillus*-dominant in cervical cancer. Compared with healthy women and women with dysplasia, many of the women with cervical cancer had more *Porphyromonas* species (Figure 4 and Appendix A).

To identify significant differences in relative abundance at the species and genus level, we further analyzed taxa abundance using a box plot (Figure 5). *Lactobacillus* predominated in the dysplasia and healthy groups and *Porphyromonas*, *Prevotella*, *Bacteroides*, and *Anaerococcus* predominated in the cervical cancer group (*p* < 0.05, Kruskal-Wallis test). At the species level, *Lactobacillus iners* predominated in the dysplasia group (*p* < 0.05) and *Porphyromonas somerae, Porphyromonas asaccharolytica*, and *Prevotella timonensis* were the most abundant species in the cervical cancer group (*p* < 0.05; Figure 5).

## 4. Discussion

Although persistent infection with high-risk HPV is a well-established risk factor and pre-requisite for cervical cancer, HPV infection is a heterogenous condition with varying outcomes, and the specific impact of other co-factors in cervical carcinogenesis is not yet well identified and characterized. Because the cervicovaginal microbiota differs with geography, ethnicity, and lifestyle, as well as infectious history, it is crucial to characterize the diversity and composition of the microbiota in different populations and the role of the microbiota in the progression of cervical cancer.

Recent literature has suggested that the cervicovaginal microbiota plays a mechanistic role in both HPV persistence and cervical cancer progression [31]. In most healthy women of reproductive age, the cervicovaginal microbiota is dominated by *Lactobacillus species*, and a lack of this dominance is recognized as a cause of adverse reproductive health outcomes [32]. Consistent with these findings, in the current study, we observed a positive relationship between alpha diversity and age (Figure 1A), as well as *Lactobacillus* dominance in the reproductive years (i.e., women younger than 50 years; Figure 1B). However, among the women with cervical cancer, there was no difference in the diversity or composition of the microbiota by age (Appendix A).

The current study was conducted among Ethiopian women with cervical cancer, histologically or cytologically confirmed dysplasia, or no evidence of cancer or dysplasia (healthy women) to characterize the diversity and composition of the cervicovaginal microbiota. To the best of our knowledge, the current study is the first to examine the relationship between the cervicovaginal microbiota and cervical cancer and/or HPV infection among women in Ethiopia. In our study, the cervicovaginal microbiota of most of the healthy women was dominated by *Lactobacillus*. Alpha and beta diversity was compared between HPV-positive and HPV-negative women regardless of their cervical cancer or dysplasia status, and we observed significant differences in beta diversity between HPV-positive and HPV-negative women, whereas alpha diversity analysis did not differ by HPV status (Appendix A).

Because our study participants with cervical cancer had different histologic diagnosis (i.e., squamous cell carcinoma or adenocarcinoma), microbial diversity was observed within the group, suggesting that microbiota diversity and composition may vary by specific cervical cancer diagnosis. Furthermore, genus and species diversity was higher in women with cervical cancer than in healthy women, which suggest that the stable composition of cervicovaginal flora, mainly *Lactobacillus* supplemented by other bacteria, is destroyed in carcinogenesis, resulting in increased microbial diversity. This finding is supported by other similar studies [17,33].

As indicated from previous studies [12,13,14], depletion of *Lactobacillus* species from the cervicovaginal microbiota structure leads to proliferation of other pathogens and hence a change of the microbiota composition, including diversity and relative abundance of genera. This explains the higher alpha diversities in the women with cervical cancer in our study compared with those with dysplasia or healthy women (Figure 2A).

Furthermore, among our study groups, relative abundance of some bacterial genera was significantly different. The predominant genus in the healthy group and dysplasia group was *Lactobacillus* (Figure 3 and Figure 4). In the cervical cancer group, the genus *Lactobacillus* decreased in relative abundance, and other bacteria, such as *Porphyromonas*, *Prevotella*, and *Anaerococcus,* were the dominant genera (Figure 5). These results are consistent with those of other studies [17,34,35]. For example, Wu et al. demonstrated less *Lactobacillus* and higher diversity of microbiota were associated with more severe pathological status. Furthermore, *Porphyromonas* and *Prevotella* were identified as cervical cancer marker genera.

The role of *Porphyromonas* and *Prevotella* in carcinogenesis has been demonstrated in oral cancers [36], with three proposed mechanisms of action: chronic inflammation, anti-apoptotic activity, and carcinogenic metabolites released by these microbes. These bacteria produce inflammatory mediators that facilitate cell proliferation, mutagenesis, oncogene activation, and angiogenesis. *Porphyromonas gingivalis*, found in oral cavities, has been reported to induce lipopolysaccharides that lead to the production of proinflammatory cytokines such as tumor necrosis factor α (TNF-α) by macrophages and interleukin (IL)-1β and IL-6 by CD4+ T helper cells [37]. Studies have also indicated that *Porphyromonas gingivalis* can mediate different signaling pathways that influence cell invasion, the cell cycle, anti-apoptosis, and inflammation [36].

Increased relative abundance of *Prevotella* in human mucosal sites has been shown to be related to various inflammatory disorders. This bacterium was indicated as a major modulator of host inflammatory responses in the female genital tract by increasing the amount of innate cytokines (IL-1α, IL-1β, IL-8, and TNFα) in cervicovaginal fluid, and production of cytokines related to Th17 (IL-23 and IL-17) and Th1 (IL-12p70 and interferon γ) [38].

The increased relative abundance of *Porphyromonas*, *Prevotella*, and *Anaerococcus* in the cervical cancer group in our study indicates that these bacteria may play a substantial role in the development of cervical cancer, supporting the previously proposed mechanisms of chronic inflammation, anti-apoptotic activity, and production of carcinogenic substances. Future studies should assesses the mechanistic relationship of a diverse cervicovaginal microbiota with cervical cancer in women in resource- limited settings, as well as the impacts of different intervention strategies. Currently, there are several treatment outlooks for cervical cancer: including the use of immunotherapy and target therapy, in combination with conventional chemotherapy or in combination with radiotherapy. Therefore, from such studies, cervicovaginal microbiota-derived bacterial markers can be used as a predictive model to predict the progression or regression of precancerous lesions and undertake further research with large sample size and possibility of identifying the right probiotics in women with persistent HPV infection or precancerous lesions of different stages, and invasive cancer that can affect chemotherapy and radiotherapy outcomes [39].

The current study had some limitations. Two different sample collection devices were used (the Evalyn brush and the Isohelix swab). Although studies have shown that the Evalyn self-sampling device performed equally well compared with samples collected by a clinician according to Illumina MiSeq sequencing of the 16S rRNA gene [40,41], the different sampling strategies may not yield comparable cervicovaginal microbiota composition and diversity. In addition, we did not consider the detailed sexual, behavioral, and clinical characteristics of the study participants. Furthermore, we used cytology, not histology, for the classification of dysplasia.

## 5. Conclusions

In conclusion, our study showed differences in cervicovaginal microbiota diversity, composition, and relative abundance between women with cervical cancer, women with dysplasia, and healthy women. The diversity and composition of the cervicovaginal microbiota increased from dysplasia to cancer, and increased levels of *L. iners* were found in women with dysplasia compared with healthy women. Differences in methods of sample collection may account for observed differences in composition and diversity. Other studies are needed to further validate the role of the cervicovaginal microbiota in development of cervical cancer.

## Figures and Tables

**Figure 1 microorganisms-11-00833-f001:**
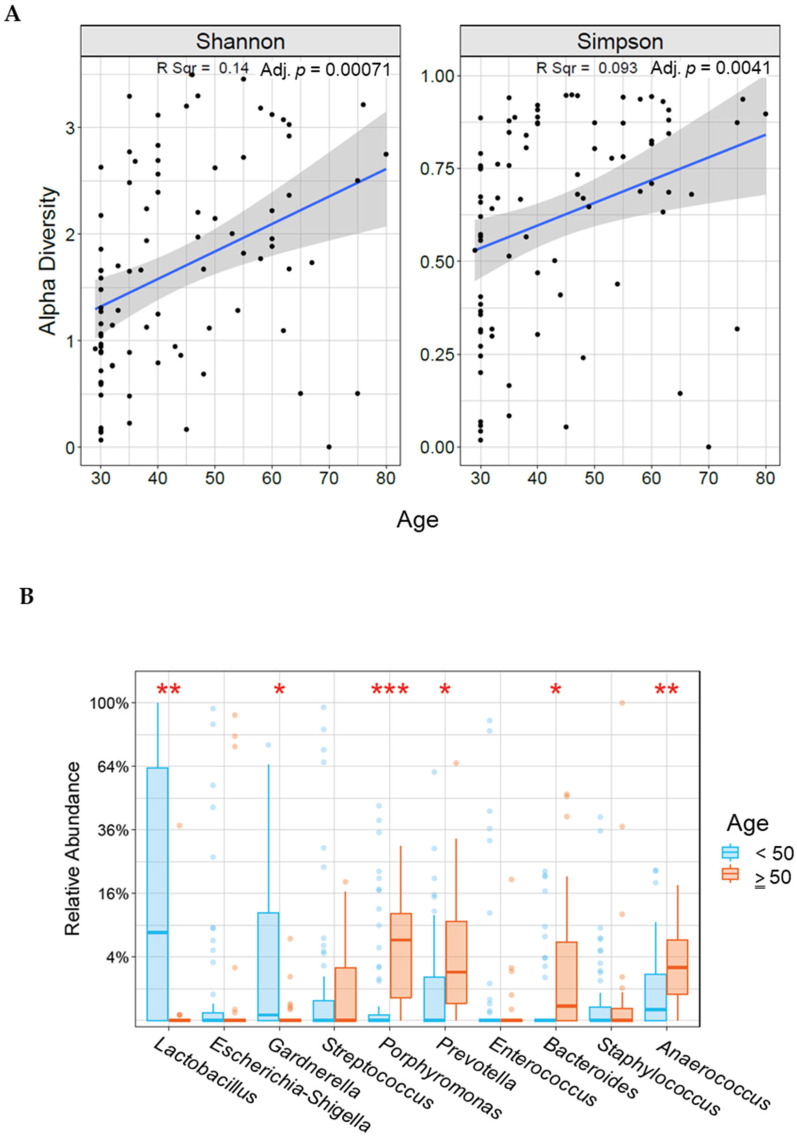
Correlation between alpha diversity (**A**) or relative abundance (**B**) and age in our study population. * indicates statistically significant difference between age groups (* <0.05, ** <0.005, *** <0.0005).

**Figure 2 microorganisms-11-00833-f002:**
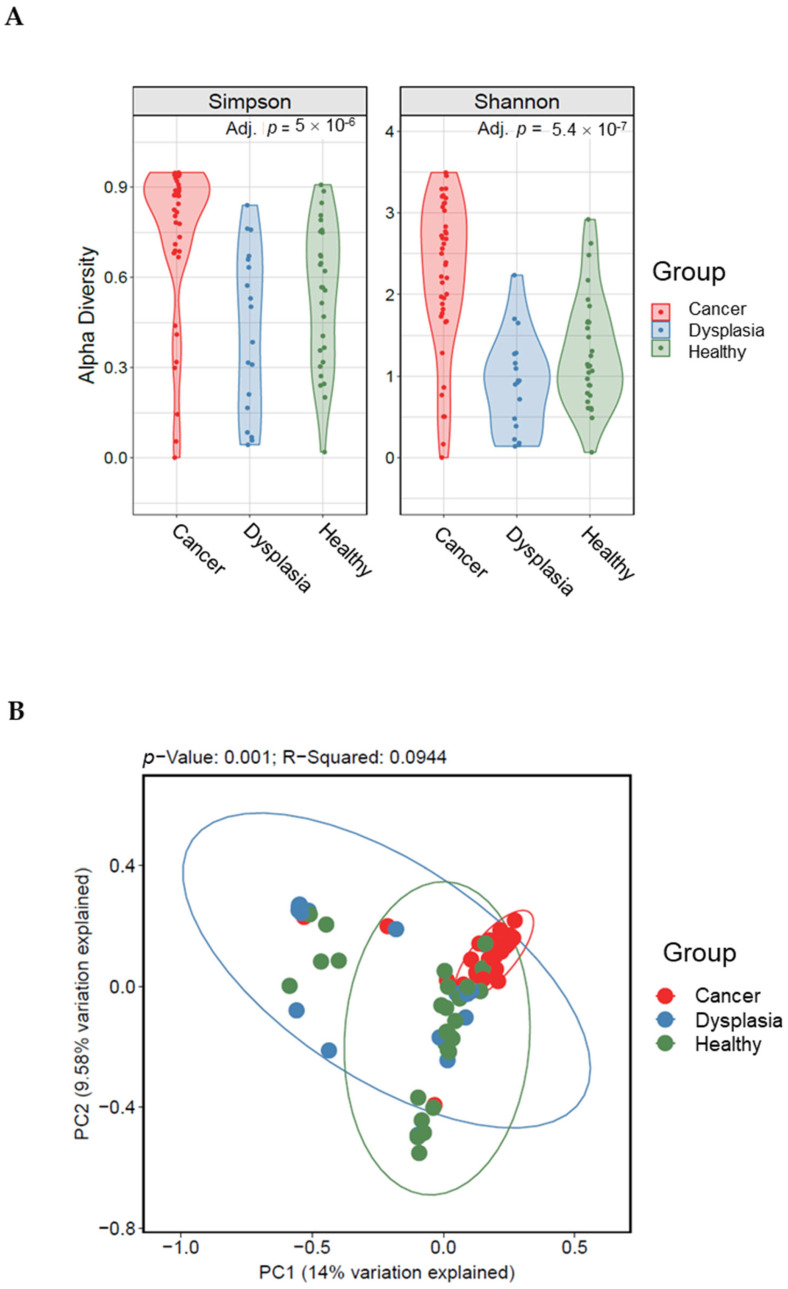
Alpha (**A**) and beta (**B**) diversity indices among women with cervical cancer, women with dysplasia, and healthy women. Beta diversity comparisons were calculated using the principal coordinate analysis discriminate. Abundance profiles in women with cervical cancer (n = 60) were different from those of women with dysplasia (n = 25) and healthy women (n = 35).

**Figure 3 microorganisms-11-00833-f003:**
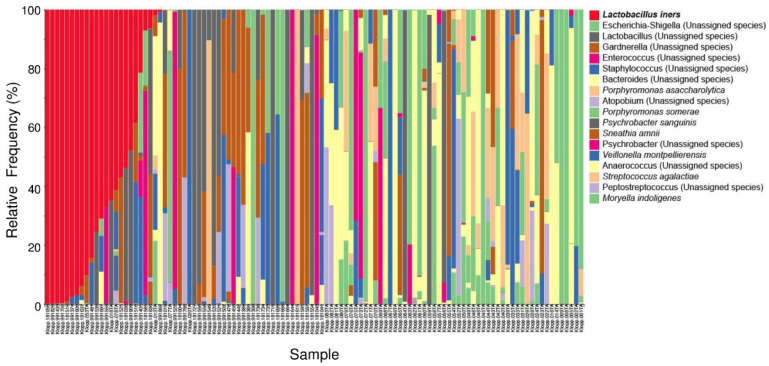
Stacked bar plot of the top 10 most abundant genus-level bacteria found in our study population. Each bar represents a single participant.

**Figure 4 microorganisms-11-00833-f004:**
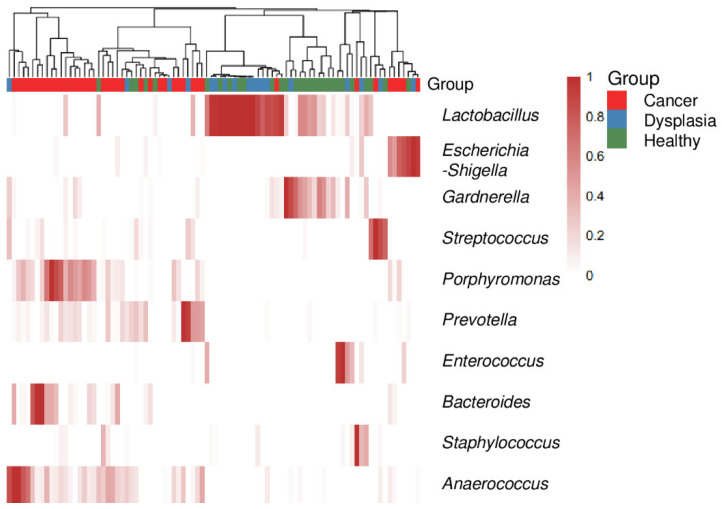
Heatmap of the top 10 most abundant genus-level bacteria in women with cervical cancer, women with dysplasia, and health women. Each bar represents a single participant.

**Figure 5 microorganisms-11-00833-f005:**
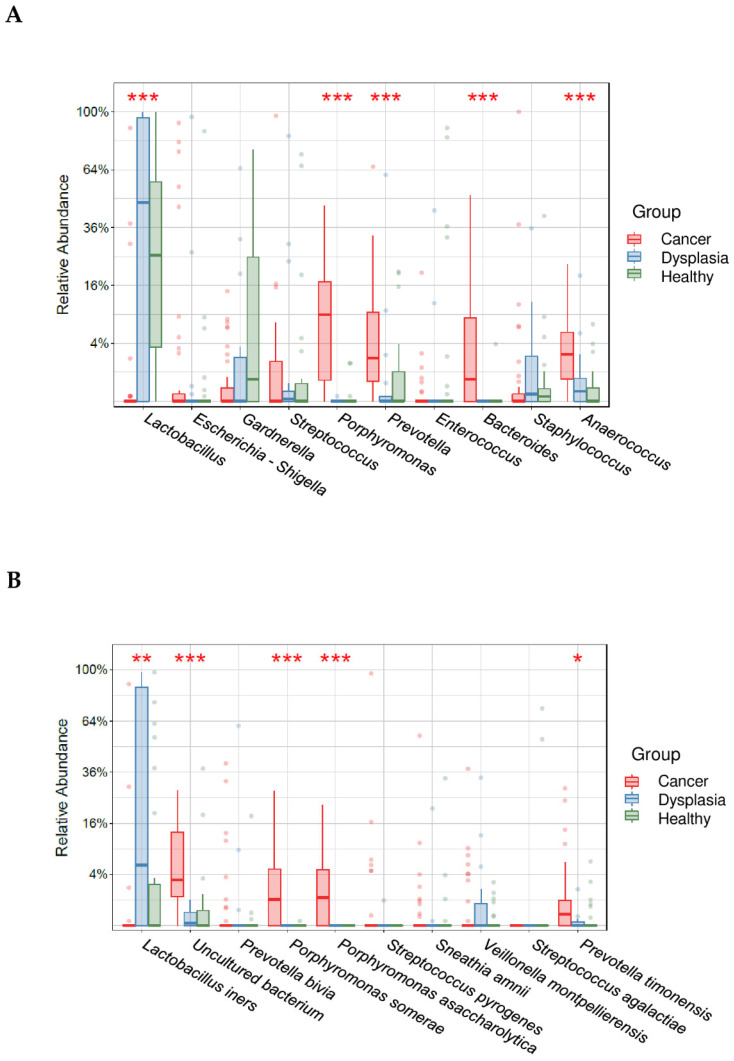
Relative abundance of the top 10 most abundant genera (**A**) and species (**B**) in women with cervical cancer, women with dysplasia, and healthy women. * indicates statistically significant difference between age groups (* <0.05, ** <0.005, *** <0.0005).

**Table 1 microorganisms-11-00833-t001:** Sociodemographic and clinical characteristics of participants (n = 120).

Characteristic	No. (%)
Age	
<50 years	84 (70)
≥50 years	35 (29)
Unknown	1 (1)
High-risk HPV infection	
Negative	27 (23)
Positive	93 (78)
HPV16 only	44 (37)
HPV18 only	2 (2)
Single HPV, not HPV16/18	20 (17)
Multiple HPV, including HPV16	20 (17)
Multiple HPV, including HPV18	3 (3)
Multiple HPV, not including HPV16/18	4 (3)
Histologic/cytologic characteristics	
ASCUS/NILM	35 (29)
Low-grade dysplasia (CIN1/2 or LGSIL)	13 (11)
High-grade dysplasia (CIN3 or HGSIL)	12 (10)
Cancer (SCC or ACC)	60 (50)

Abbreviations: HPV, human papillomavirus; ASCUS, atypical squamous cells of undetermined significance; NILM, negative for intraepithelial lesion or malignancy; CIN, cervical intraepithelial neoplasia; LGSIL, low-grade squamous epithelial lesion; HGSIL, high-grade squamous epithelial lesion; SCC, squamous cell carcinoma; ACC, adenocarcinoma.

## Data Availability

Data are available upon reasonable request. The data supporting this study’s findings are available from the corresponding author, [BT], upon reasonable request.

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
