# Peer review of "Cervicovaginal Microbiota Profiles in Precancerous Lesions and Cervical Cancer among Ethiopian Women"

_microorganisms, 2023, doi:10.3390/microorganisms11040833_

Round 1

Reviewer 1 Report

This study descibes the association of cervicovaginal microbiota profiles with precancer lesions and cervical cancer. It is very important to investigate the role of candidate bacteria in the progression of cancer. Authors can further explore the effect of candidate bacteria on growth (proliferation), migration and invasion in cancer cells. At least, authors should provide more information about that how do these candidate bacteria influence the development of cancer in the section of Discussion.

English language and style needs to be checked.

Reviewer 2 Report

The main goal of the undertaken research was the analysis of the cervicovaginal microbiota

 in women with cervical cancer and/or HPV positive and HPV negative. 

The authors emphasize that to the best of their knowledge, this study is  the first in Ethiopia. 

The types of Porphyromonas, Prevotella and Anaerococcus dominated in women with cancer, unlike women from the control group where Lactobacillus was most often detected.   The research was conducted reliably. The selection of groups for the study was described in detail (Inclusion and exclusion criteria).  The graphical presentation of test results is very clear. My comments on the Discussion section are following:

1.     It would be better if the authors wrote whether similar dependencies exist in other parts of the world. 

2.     How Porhyromonas and HPV co-infection affects cencerogenesis?

Reviewer 3 Report

Dear Authors

This manuscript investigated the cervicovaginal microbiota profiles in cervical cancer, cervical dysplasia, and healthy women and verified the difference in cervical microbiota diversity, composition, and relative abundance between cervical cancer patients, patients with dysplasia, and controls. The authors also showed no difference in alpha, beta diversities, and microbiota in the cancer group between <50 years and > 50 years because the two age groups have an uneven distribution of histology/cytology characteristics.

However, it could not be verified if cervical microbiota diversity, composition, and relative abundance between cervical cancer patients, patients with dysplasia, and controls had differences in the >50 years group because age-associated alterations in cervical microbiota composition and diversity were observed, and there were few dysplasia patients and healthy women who were >50 years. Therefore, it will be better to limit the author’s conclusion to the reproductive ages. It is also important to discuss the reason why the number of dysplasia patients and healthy women who were >50 years was few.

There were several other points to be revised as shown below.

1.      In Abstract, Line 36-38, where is the data?

2.      In Results, show the median age of subjects.

3.      In Discussion, Line 299-301, what does it mean?

4.      In Conclusion, Line 357, where is the data of ‘increased L. iners was found in women with dysplasia’?

Reviewer 4 Report

Authors presented a study to investigate the effects of cervicovaginal microbiome on cervical lesions.

In my opinion, the topic is interesting enough to attract the readers’ attention; however, some points may be improved.

- The whole text should be corrected by a native English speaker in order to make the work clearer and more readable. 

-Typos errors should be corrected.

-The introduction should be extended and completed. I find interesting a reference to the efforts made for the prevention and early diagnosis of gynecological cancers (see PMID: 36141217).

- What is already known on this subject? What do the results of this study add? 

-What are the implications of these findings for clinical practice and/or further research?

- Discussions can be expanded and improved by citing relevant articles (I suggest authors to read and insert in references the following article PMID: 35742340).

Considered all these points, I think it could be of interest for the readers and, in my opinion, it deserves the priority to be published after major revisions.

Round 2

Reviewer 3 Report

The authors mostly revised the paper according to the reviewers' comments, but they were unable to explain the age bias.

Author Response

Dear Reviewer, thank you very much for your concern regarding the age bias. As you well stated, the cervicovaginal microbiota diversity and composition may vary with age. Therefore, the uneven distribution of histologic or cytologic characteristics among the two age groups in our study was the limitation of the study. 

Thank you.

Reviewer 4 Report

The introductory section has been expanded with recommended reference (PMID: 36141217), but this has not been reflected in the references section. Discussion section  has not been expanded as suggested. (PMID: 35742340) Please make sure that these references are also added in the "References" section according to the numerical order of the manuscript text.. 

I suggest improving the quality of the manuscript because after these modifications,  the manuscript can be accepted.

Author Response

Dear Reviewer, thank you once again for your important recommendation. I am very sorry for not included the recommended article in the reference list although cited in the introduction part.

Our discussion was now expanded according to your recommendation using the relevant references (see lines 450-457). 

Thank you.